# Validation of Novel Ultrasonic Phased Array Borehole Probe by Using Simulation and Measurement

**DOI:** 10.3390/s22249823

**Published:** 2022-12-14

**Authors:** Prathik Prabhakara, Frank Mielentz, Heiko Stolpe, Matthias Behrens, Vera Lay, Ernst Niederleithinger

**Affiliations:** Bundesanstalt für Materialforschung und -Prüfung (BAM), 12205 Berlin, Germany

**Keywords:** borehole probe, engineered barrier, monitoring, phased array technique, ultrasonic testing, non-destructive testing civil engineering

## Abstract

Low-frequency ultrasonic testing is a well-established non-destructive testing (NDT) method in civil engineering for material characterization and the localization of cracks, reinforcing bars and delamination. A novel ultrasonic borehole probe is developed for in situ quality assurance of sealing structures in radioactive waste repositories using existing research boreholes. The aim is to examine the sealing structures made of salt concrete for any possible cracks and delamination and to localize built-in components. A prototype has been developed using 12 individual horizontal dry point contact (DPC) shear wave transducers separated by equidistant transmitter/receiver arrays. The probe is equipped with a commercially available portable ultrasonic flaw detector used in the NDT civil engineering industry. To increase the sound pressure generated, the number of transducers in the novel probe is increased to 32 transducers. In addition, the timed excitation of each transducer directs a focused beam of sound to a specific angle and distance based on the previously calculated delay time. This narrows the sensitivity of test volume and improves the signal-to-noise ratio of the received signals. In this paper, the newly designed phased array borehole probe is validated by beam computation in the CIVA software and experimental investigations on a half-cylindrical test specimen to investigate the directional characteristics. In combination with geophysical reconstruction methods, it is expected that an optimised radiation pattern of the probe will improve the signal quality and thus increase the reliability of the imaging results. This is an important consideration for the construction of safe sealing structures for the safe disposal of radioactive or toxic waste.

## 1. Introduction

The ultrasonic pulse-echo method is an established technique in civil engineering for the continuous evaluation of concrete structures [1,2]. The embedded sensors and various data processing techniques are used to develop applications such as structural health monitoring and quality assurance related to structural integrity. In concrete structures, ultrasonic testing is used to detect potential defects, locate, and monitor the condition of tendons, characterize the material, and measure the thickness of objects [1,3]. The inhomogeneous nature of concrete, such as reinforcement, aggregates of different sizes and matrices, causes structural noise. In addition, using a coupling agent on the rough surface limits the application of ultrasonic testing in civil engineering. However, recent developments, such as using a dry point contact (DPC) transducer (Figure 1) for good acoustic contact and low-frequency transducers (20–100 kHz) to minimize interaction with coarse concrete, are widely used in non-destructive testing in civil engineering [1,4].

In Germany, radioactive waste is safely isolated from the human environment and stored in stable geological formations in underground repositories operated by the federal company for radioactive waste disposal in Germany (BGE). The sealing structures are an essential element of the safe disposal of radioactive waste. For salt as a host rock, BGE runs long-term tests with sealing structures in the mine Morsleben in Germany. The existing test structure at the Morsleben site consists of a special salt concrete material. After initial construction, monitoring and inspection, research boreholes are drilled for exploration or operational purposes, while the actual barriers will have no boreholes [5,6]. In order to improve the construction material, testing procedures and quality assurance through non-destructive testing, BAM started the internally funded project SealWasteSafe (SWS) [7,8]. One of the main objectives of this project is to develop a phased array ultrasonic borehole probe that will be validated on laboratory test specimens before later testing at in situ test sealing structures.

So far, ultrasonic borehole probes have only been used in geophysics as mini-seismic to study rock mechanics, geology, and geotechnical site investigation in the Mont Terri rock laboratory [9]. The study conducted in [9] uses elastic waves with refraction, transmission, and partially reflected waves to correlate and interpret seismically derived parameters with relevant rock mechanics parameters.

The borehole probe is a pulse-echo ultrasonic device used to investigate the internal structure of a sealing system using existing research boreholes [10]. The probe consists of a series of individual coupling agent-free shear horizontal wave DPC transducers (Figure 1) operating at a frequency of 50 kHz. During the measurement, the probe is inserted into the desired depth of the borehole and pneumatically coupled with air pressure (Figure 2a). Inside the borehole, the measurement is carried out in different directions along the length of the borehole. A prototype borehole probe was developed with an array of 12 individual horizontal shear wave transducers polarized perpendicular to the axis of the array and operating at a frequency of 50 kHz (Figure 2b) [10]. The array consists of a combination of six transmitters and six receivers operated with a commercially available flaw detector and an external amplifier. The measured data is then analyzed using SAFT (Synthetic Aperture Focusing Technique) to locate cracks, delamination, and built-in parts [3,11].

To increase the generated sound pressure and to adapt the phased array technique, a second-generation borehole probe was developed consisting of 16 shear wave transducers. Each array of transducers in the second generation borehole probe consists of eight DPC transducers that act separately as transmitters and receivers (Figure 3a) [6]. So far, the second generation borehole probe is used for ultrasonic investigation in sealing structures without using the phased array technique, penetrating up to 1 m deep. The problem with using a conventional unfocused beam in the sealing structure is if there is a salt-to-salt transition or greater heterogeneity, which can cause structural noise that masks the defect response. A dipping reflector (the crack inclined more than 30 degrees) cannot be reconstructed with the existing method.

The borehole probe was redesigned as a third generation to further increase the sound pressure level with an integrated phased array technique and a built-in amplifier in the assembly of the probe to improve the signal amplitude. The third generation borehole probe developed for the SWS project consists of 32 DPC transducers operating at a frequency of 50 kHz (Figure 3b). Each row of the array consists of 16 transducers operating separately as transmitter and receiver, allowing a depth penetration of the sound field of up to 2 m. The constructed novel phased array probe is intended to improve the signal-to-noise ratio, which increases the reliability of the imaging results, and to be suitable for sizing the cracks by directing the beam in a specific direction.

In this paper, preliminary testing was carried out on a hemispherical polyamide sample with a point transducer to validate the radiation pattern of the horizontal shear wave in two planes of polarization. A similar procedure is then applied to the second generation borehole probe by adapting the phased array technique to investigate the directivity of different focusing angles based on the calculated delay time and comparing the results with the simulation. Finally, the test measurement with the new third generation borehole probe is investigated for the directivity on a newly built half-cylindrical concrete specimen and compared with the simulation.

## 2. Materials and Methods

### 2.1. Phased Array Technique

Phased array technology or phased array ultrasonic testing (PAUT) is an advanced non-destructive testing technique that uses a series of transducers or elements. Each of these transducers is individually pulsed with a calculated delay time to focus and sweep the beam inside the test object without moving the probe. This feature of the phased array method can improve the effect of high attenuation due to the anisotropic structure of concrete material on the response signals and increases the signal-to-noise ratio [12]. The development of the borehole probe with a dynamic adaptation of the probe aperture and focussing of the sound field will be carried out using the phased array technique. With the help of the excitation function, which means the delay time of the individual transducers, the sound field can be focused and swivelled to a required angle depending on the calculated delay time [13]. This limits the recorded test volume and improves the signal-to-noise ratio of the received signal. Figure 4 shows a schematic representation of the possible time-controlled ultrasound phased array technique [14].

Currently, the control of the probe and the data acquisition are not yet integrated into the housing of the borehole probe. The electronics are already designed for future developments of the integrated phased array system in the third generation borehole probe. Figure 5 shows the block diagram of a phased array measurement system with a borehole probe [10].

A new type of multi-channel high-voltage transmitter having a peak to peak voltage of ± 170 V and a duration of 4 µS to 40 µS (50 kHz => period (Ti) = 20 µS) with bipolar square-wave signals was developed for the excitation of probes with programmed time delays (Figure 6). This multi-channel transmitter is operated with the help of a LabVIEW program that excites the individual transducers based on the calculated delay time.

### 2.2. Shear Horizontal (SH) Wave

The horizontal shear wave refers to the particle motion of the wave, which means the displacement and velocities in the different directions of the wave propagation. To obtain the radiation pattern of the SH wave, the magnitude of the velocity vector at each data point is calculated and the maximum amplitude for each azimuth is determined. This procedure is similar to the work of Maack [15], who selects the maximum amplitude from the recorded signal on the hemispherical specimen. To compare the results with the analytical solution, an approximation (Equation (1)) by Kutzner [16] is used. Figure 7 shows the polarization of the wave in two different planes of radiation pattern calculated by using Equation (1) [15,16].
(1)$$b_{AT}\left(\alpha \right)=\frac{\left(\;1-2sin^{2}\alpha \;\right)cos\alpha \;}{{\left(1-2sin^{2}\alpha \right)}^{2}+4sin^{2}\alpha cos\alpha \sqrt{k^{2}-sin^{2}\alpha }}$$

▪Velocity ratio, $k=\frac{C_{T\left(polyamide\right)}}{C_{L\left(polyamide\right)}}$▪$C_{T\left(polyamide\right)}=1121\;m/s$—Transverse wave velocity in polyamide▪$C_{L\left(polyamide\right)}=2642\;m/s$—Longitudinal wave velocity in polyamide▪α azimuth angles vary between −90° to +90

Here $b_{AT}\left(\alpha \right)$ is the radiation pattern of a point SH wave in the far field. The angle α is measured in the direction normal to the surface on which the source acts *(|α| ≤ π/2)*. Since the velocity of the longitudinal wave is greater than that of the transverse wave *(CT ≤ CL)*, the equation becomes more complex for larger angles. Therefore, *|b(α)|* is considered in the representation of the wave amplitude dependence in the subsurface [15,16].

### 2.3. Directivity Measurement

In order to investigate the directivity of the wave field, a similar approach to that used by Maack [15] is followed, using a polyamide half sphere with a diameter of 587 mm as a reference test specimen [15]. According to the definition, directivity contains two measurand; on the one hand, the spatial orientation of the measuring point by its coordinates, and on the other hand, the amplitude of the sound pressure. No material influences are considered when determining a sound field’s directivity. An automatic scanner is developed and built to measure the spatial wave field inside the homogeneous polyamide test specimen (Figure 8) [15]. A transmitter is mounted on the flat surface of the hemispherical test specimen, which is freely accessible from the lower circular disc. The receiver is attached to a frame-like swivel arm, which is movably attached to the base frame of the test stand and is operated with the LabView program. With this automatic ball scanner, a preliminary study is carried out to measure the radiation pattern of the SH point transducer. The maximum amplitude from all measurement points is plotted in polar coordinates to visualize the transducer’s radiation pattern. A similar measurement is performed on this homogeneous polyamide test specimen with a second generation probe to investigate the directivity pattern of the phased array technique.

The newly constructed third generation borehole probe (Figure 3) with a larger aperture is validated using a new half-cylindrical test specimen. This new specimen was manufactured from concrete having a half-cylindrical shape with a radius of 750 mm and a length of 800 mm without any reinforcement (Figure 9a). The construction of the test specimens complies with the DIN 18551 standard with strength class C30/37 and a curing time of 28 days (“C” refers to the compressive strength, 30 MPa/28 days for the cylinder test and 37 MPa/28 days for the cube test) and a maximum grain size of 8 mm. A fixture was made to attach the borehole probe to the flat surface of this half-cylindrical test specimen (Figure 9b). A single-point receiver was mounted on a circular disc and moved manually on the test surface and oriented as shown in Figure 9c. With the constant air pressure, the receiver is fixed at the different points on the circular surface while taking measurements.

### 2.4. Modelling and Simulation

A finite element method with a semi-analytical approach is implemented in the CIVA software developed by the French Alternative Energies and Atomic Energy Commission (CEA) [17]. CIVA is widely used in industrial applications, as it is known for its efficient approach to modelling and simulation of inspection problems. The module for ultrasonic testing in CIVA consists of beam computation and inspection simulation. The beam calculation is mainly used to investigate the propagation of ultrasonic waves. We use the beam calculation to validate the phased array system of the new borehole probe. The first essential parameters are the test specimen geometry and material properties (density and shear wave velocity). A borehole probe is designed using a point source transducer with 16 element transmitters with a 20 mm distance between each element and polarization perpendicular to the test plane. The signal parameters obtained from the transducer data sheet and the array settings are used to calculate the delay time for deflecting the beam to the desired angle. This forward calculation was carried out successfully [18].

### 2.5. Delay Time Calculation

In phased array technology, the most important parameter is the delay law or delay time for each array element that sweeps and focuses the beam. The delay time is automatically calculated in the CIVA software by defining parameters such as focusing distance and sweep angle. To verify the delay time calculated by the software, let us consider a focus point “*P*” located at a distance “*F*” from the centre of the array (diffraction is neglected in the calculation) and having an angle “*θ*” (Figure 10). The delay time is calculated using Equation (2) [18], that is, the largest travel time from the first element to the point “*P*” is subtracted from the individual travel times of the various elements. In the diagram, the unknowns *R*_1_ and *R_n_* are calculated by applying the cosine rule to the triangle [19,20].
(2)$$Delay\;time\;(∆t_{n})=\frac{R_{1}-R_{n}}{V}$$
(3)$$R_{n}=\sqrt{F^{2}+e_{n}{}^{2}-2\times F\times e_{n}sin\theta }$$
(4)$$R_{1}=\sqrt{F^{2}+{\left(\frac{\left(N-1\right)\times a}{2}\right)}^{2}+2\times F\times \left(\frac{\left(N-1\right)\times a}{2}\right)\times sin\theta }$$
*F*—Focusing distance from center array; *θ*—Focusing angle; *P*—Point focus; *a*—Distance between individual transducer; *N*—Number of transducers; *en*—Length of half aperture; *V*—Shear wave velocity.

### 2.6. Data Acquisition

Data acquisition for the directivity measurement is illustrated in the block diagram (Figure 11). The borehole probe is attached to the lower flat surface of the half-cylindrical specimen using pneumatic air pressure. The probe with one to 16 transmit channels is connected to a multi-channel high voltage transmitter (Figure 6b) and a NI-PXI multiplexer (16 differential/32 single-ended channels, nominal input ranges ± 0.2 V to ± 10 V for ADC or DAC). With the NI-PXI-1042 controller, the calculated delay time for each channel is defined using a LabView program. A single DPC SH transducer guided by a wooden mount serves as a receiver on the upper circular surface of the half-cylindrical specimen, which is connected to an ultrasonic band amplifier (20 kHz to 300 kHz at −3 dB, +5 V) to amplify the signal. Data acquisition is made using the LabView program on an external laptop. During the measurement, the transducer is moved manually from 10° to 170° with a step of 1° using pneumatic air pressure. All measurement data is stored in a single file without headers and processed with a Python program. A band-pass filter (10 kHz to 60 kHz) is applied when processing the measurement data. The Hilbert envelope determines the maximum amplitude of each measurement position displayed in polar coordinates to see the radiation pattern.

## 3. Results and Discussion

### 3.1. Directivity Measurement of SH Point Source

Measurements are performed on a hemispherical polyamide sample to determine the source characteristics of the horizontal shear wave. The point source transmitter mounted on the flat base transmits the signal inside the polyamide sample (Figure 12a) and receives the signal on the circular top from 10° to 170° in a step of 1° with an automatic scanner controlled by a LabVIEW program (Figure 12b). During acquisition, the amplifier at the receiver point source is used to amplify the signal. The shear wave polarization is measured by rotating the circular disc by 90° and repeating the same measurement.

Figure 13 shows the filtered signal and the Hilbert envelope of the signal received at 10° on the circular surface of the polyamide specimen. The maximum envelope amplitudes of each received signal are plotted in polar coordinates with respect to the azimuth angle to obtain the radiation pattern. Figure 14 shows the radiation pattern of the point source measured in the direction of the receiver and another one perpendicular to the plane in which the point source is rotated by 90°. The simulation is carried out with the CIVA UT module to compare the results with the measurement. The known material properties of polyamide, that is, shear wave velocity = 1121 m/s, longitudinal wave velocity = 2642 m/s and density = 1.14 gm/cm^3^, and a point source with a center frequency of 50 kHz, are defined. Figure 15 shows the simulation of the point source in parallel and perpendicular planes corresponding to the two main polarization directions.

Figure 16 shows the direct comparison of the directional characteristics of calculation, simulation, and measurement. In the left figure, the main lobe, and the side lobe of the sound field of a single point source are clearly visible for shear wave polarized along the plane. In the measured directional characteristics (Figure 15a, blue color), the amplitude values for the main lobe are increased by about −27° and 27° compared to the calculation. This could be due to the superposition of head waves of the shear wave, which depend on the radius of the hemispherical polyamide test specimen [15,21]. The cause of the increased amplitude needs further investigation.

Figure 16b shows the directivity of a point transducer where the emitted shear wave is polarization perpendicular to the test plane. It can be clearly seen that the results of the calculation, the simulation, and the measurement correlate very well with each other.

### 3.2. Validation of Second Generation Borehole Probe

A similar measurement is carried out with the second generation borehole probe on a polyamide test specimen to validate the phased array technique. The borehole probe was constructed with an SH-wave transducer with polarization perpendicular to the array axis, transmitting spherical waves. The second generation borehole probe shown in Figure 3 is attached to the flat bottom of the test specimen and serves as the transmitter. A multi-channel transmitter is used to apply delay time that directs the beam at specific angles inside the specimen. An example of a 45° beam shown in Figure 17 is the borehole probe directivity measurement and simulation radiation pattern. The signal is measured along the circular axis from 10° to 170° in steps of 1°. The maximum amplitude of each measurement signal (from 10° to 170°) is plotted in polar coordinates. A corresponding simulation is performed on the polyamide test specimen with a borehole probe. An eight-element point source probe is designed in CIVA with a distance of 20 mm between the individual elements. The calculated delay times are defined before measurement and simulation using an estimated mean velocity of the sample. Examples of the calculated delay times are shown in Appendix A, Table A1 for a 45° focus beam. Figure 17c shows that the measurement and simulation results show good agreement for the calculated delay time that successfully steers the beam to 45°. However, there is a difference in the side lobes at an angle of −50°, and the amplitude value is higher in the measurement than in the simulation. This could be due to the default apodizing factor for the transducer in the simulation software, which reduces the side lobes. The simulation result shows a focusing distance of up to 208 mm, which will be further analyzed in the upcoming validation tests with the new borehole probe in terms of agreement with the measurements.

### 3.3. Velocity Measurement of Concrete Test Specimen

The transmission measurement is performed on the 90°-axis of a half-cylindrical concrete specimen (Figure 9) with point shear wave transducers to determine the mean velocity with the known geometry of the specimen. The shear wave velocity is calculated by dividing the radius of the semi-cylindrical specimen (750 mm) by the first maxima of the shear wave arrival time at different points along the length of the specimen. Figure 18 shows the velocity of the concrete specimen at different measurement points with an average velocity of 2657 m/s and a standard deviation of 12.9 m/s.

This velocity parameter is an essential parameter for calculating the delay time for the phased array technique. In practice, this velocity is not fully constant, and variations of the speed are unknown. Further research will be necessary to analyze the effect of these velocity variations on the focusing.

### 3.4. Nearfield Distance Simulation of Third Generation Borehole Probe

The sound pressure in the vicinity of the wave source varies significantly due to the interference of the wave, and a series of sound pressure maxima and minima occur with harmonic excitation. The distance between the sound source and the last sound pressure maximum is known as the near-field distance [12,19]. In CIVA, the near-field of the borehole probe is simulated using harmonic simulation, taking into account the probe parameters of a 16-element array and the material properties of the concrete specimen. Figure 19 shows the beam calculation of an unfocused third generation borehole probe, resulting in a near-field distance of 0.430 m. In the near field, the sound pressure oscillates along the axis, but in the far field it decreases continuously; the point where the amplitude starts to decrease in Figure 20 is assumed to be the near field distance [13,22].

The near field distance (N) for the third generation borehole probe for the concrete test specimen is calculated numerically using Equation (6). This formula is a rough approximation and is only valid for circular transducers, which are described by Mielentz [13]. Here ‘*f*’ is the frequency, ‘*d*’ the aperture and ‘*n*’ number of elements of the borehole probe, ‘*a*’ is the distance between the individual elements, and the shear velocity (*V*) of the new concrete specimen is 2657 m/s. The simulation results agree well with the calculated near-field distance of 0.4234 m.
(5)$$Aperture\left(D\right)=\left(N-1\right)\times a=\left(16-1\right)\times 0.02\;m=0.3\;m$$
(6)$$Near\;Field\;\left(N\right)=\frac{fD^{2}}{4V}=\frac{50000\;\mathrm{Hz}\times {\left(0.3\;m\right)}^{2}}{4\times 2657\frac{m}{\sec}}=0.4234\;m$$
*f*—Centre frequency; *V*—Shear wave velocity; *D*—Aperture; *a*—Distance between individual transducer; *N*—Number of transducers.

### 3.5. Validation of Third Generation Borehole Probe

The directivity measurement is carried out with a novel third generation borehole probe on a newly built half-cylindrical concrete test specimen as per Section 2.6 data acquisition. The measurement setup is shown in Figure 21, where the probe is attached to the flat bottom surface using a bracket and pneumatic vacuum. The newly constructed multi-channel transmitter is used to assign a delay time to the individual elements. The point shear wave receiver on the upper circular surface is mechanically moved to take measurements from 10° to 170° in a step of 1°. The measurement is performed for different focusing angles (0°, 20°, 45°, and 60°); the result shows (Figure 22a) the measurement of the 45⁰ focused beam based on the calculated delay time (Table A2). The maximum amplitude from each signal is plotted in the polar coordinates to visualize the radiation pattern of the measurement. The homogeneous simulation is performed by modelling the 16-array probe and considering the signal and material parameters (velocity = 2657 m/s) with the calculated delay times used in the measurement.

The measurement result shows the main lobe at 45° and some side lobes with minimum amplitude in different angular directions. This might be because of delay time on the coarse-grained structure, which has strong attenuation and twisting effects. The effect of delay time on the coarse-grained structure is one of the interesting topics to study further in phased array technology. However, the main objective of this research study is to increase the signal-to-noise ratio and improve the imaging over conventional ultrasonic inspection of concrete structures. A comparison of the measurement and simulation results (Figure 22c) shows good agreement for the calculated delay time, which successfully steers the beam to 45°. However, the width of the beam is significantly smaller than in the simulation, which cannot be explained without further investigation.

## 4. Conclusions

The preliminary study with measurement and simulation to validate the principle of the horizontal point transducer agrees well with the theoretical radiation pattern. For the second generation borehole probe consisting of 16 transducers, the beam computation with the calculated delay times correlates well with the measurement results on polyamide test specimens for a 45° focus beam. A new 32-element phased array borehole probe was designed and built to direct and focus the beam to the desired angle based on the calculated delay time. The main objective of this new borehole probe is to penetrate the sound field to a greater depth, and the phased array technique is a promising method to determine the defect size, which is essential for the investigation during operation. To validate the new third generation borehole probe, a semi-cylindrical concrete specimen was built to investigate the directional characteristics. The calculated near-field distance for the new borehole probe and the simulation resulted in approx. 430 mm for the 16-element transmitter. In the last part, the first measurement of the directivity on the concrete specimen with the new probe for different focusing angles agreed well with the simulation result, so the principle was successfully validated.

Currently, the phased array technology and the amplifier to amplify the signal during the measurement are being installed in the third generation borehole probe. The probe is currently being tested with laboratory samples for further detailed investigations. In addition, once the laboratory measurements are completed, the first trial measurements with the third generation borehole probe are planned for BGE’s Morsleben site. The measurement data will then be used to work on the imaging technique with SAFT and the geophysical reconstruction method. Overall, the improved results from the borehole probes will be relevant in the coming years for high-quality investigations on large concrete structures for quality assurance and to enable safe sealing structures.

## Figures and Tables

**Figure 1 sensors-22-09823-f001:**
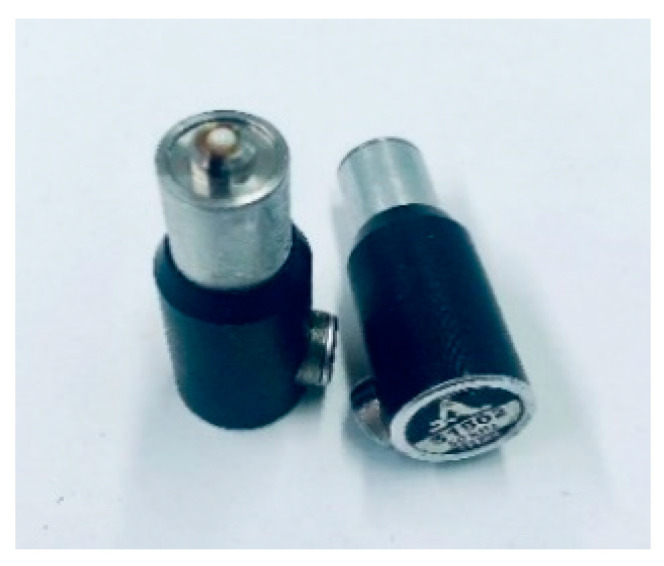
DPC Shear wave transducer.

**Figure 2 sensors-22-09823-f002:**
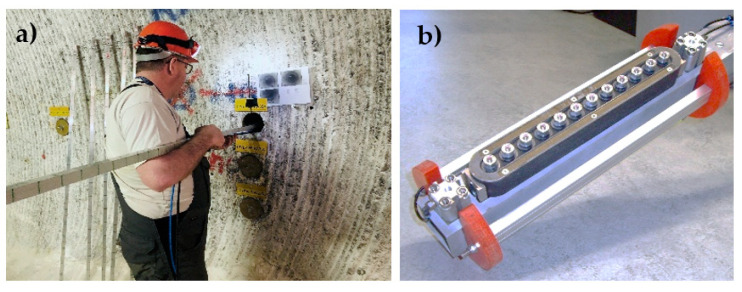
(**a**) Borehole probe measurement at sealing structure, (**b**) Prototype borehole probe.

**Figure 3 sensors-22-09823-f003:**
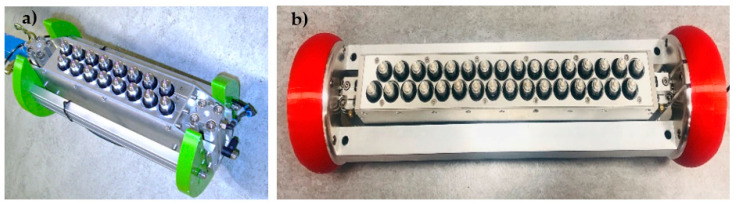
(**a**) Second generation borehole probe, (**b**) novel third generation borehole probe.

**Figure 4 sensors-22-09823-f004:**
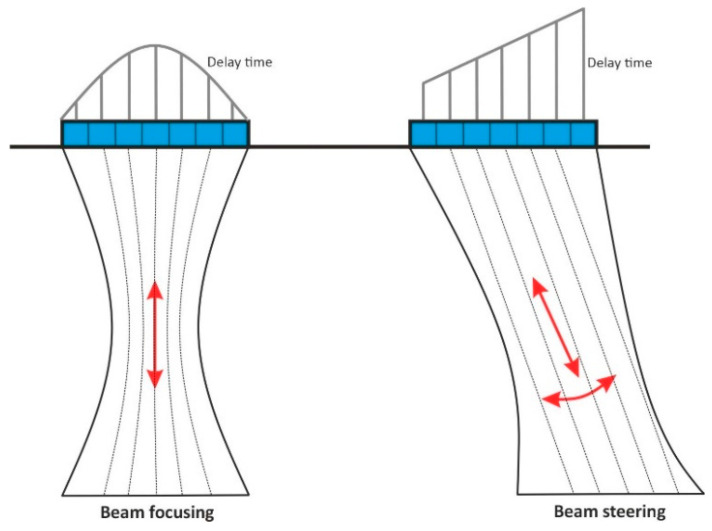
Schematic representation of phased array technique.

**Figure 5 sensors-22-09823-f005:**
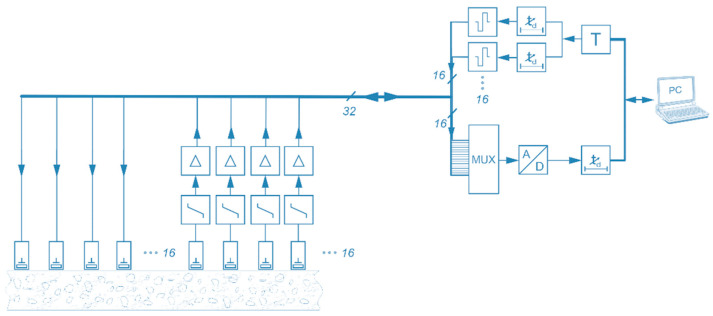
Block diagram of ultrasonic phased array system with borehole probe.

**Figure 6 sensors-22-09823-f006:**
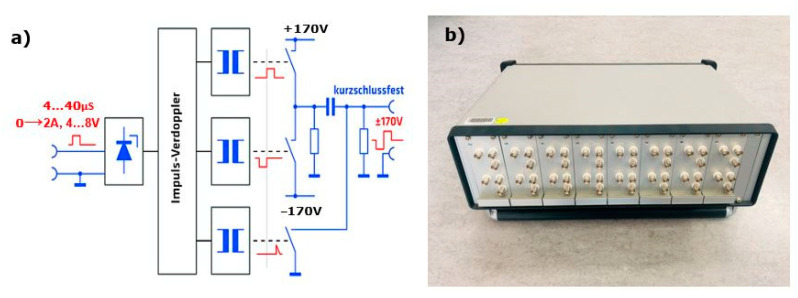
(**a**) Block diagram of single transmission stage, (**b**) Assembly of multi-channel transmitter.

**Figure 7 sensors-22-09823-f007:**
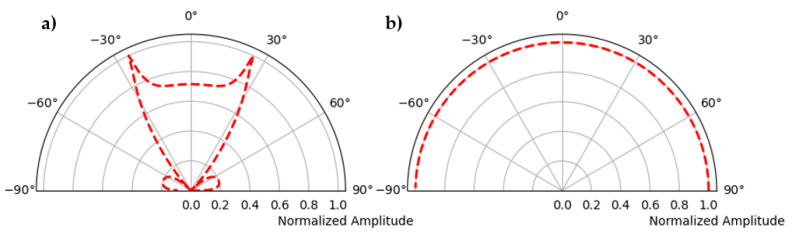
(**a**) Polarization along the test plane, (**b**) polarization perpendicular to the test plane.

**Figure 8 sensors-22-09823-f008:**
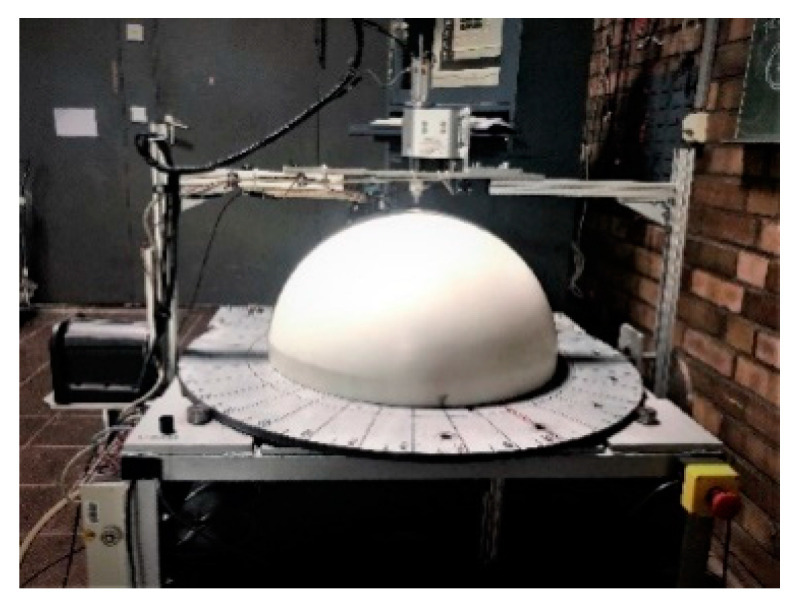
Polyamide half sphere test specimen with automatic ball scanner.

**Figure 9 sensors-22-09823-f009:**
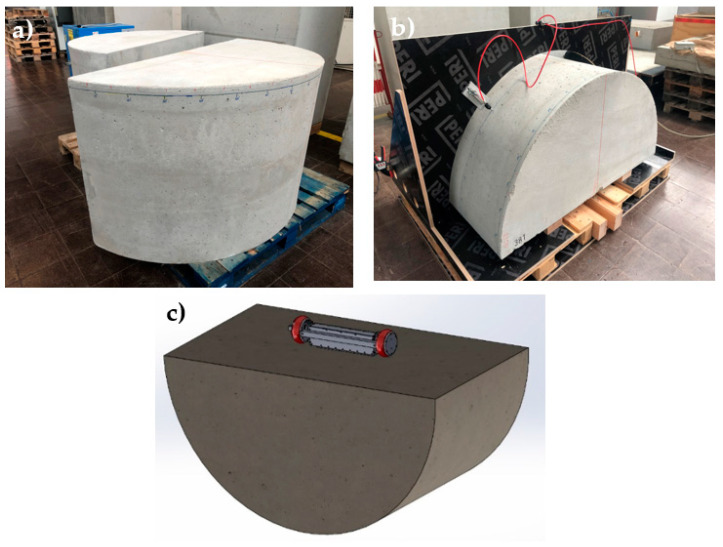
(**a**) Concrete half cylindrical test specimen, (**b**) measurement setup for directivity measurement and, (**c**) borehole probe fixed at the flat surface of the test specimen.

**Figure 10 sensors-22-09823-f010:**
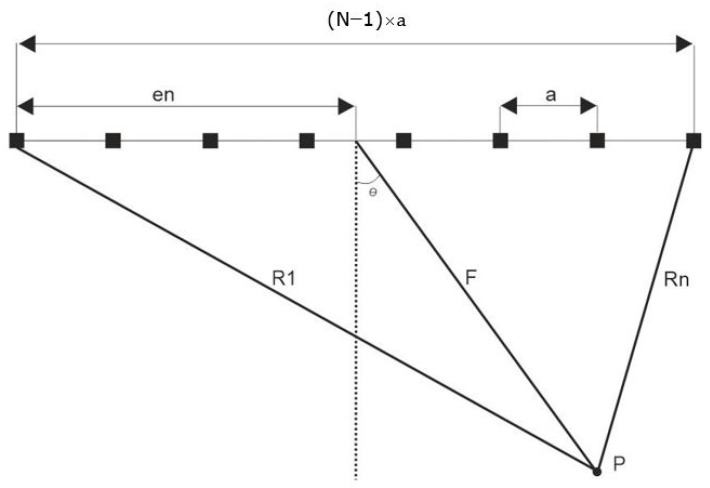
Schematic representation of delay time calculation for point focus.

**Figure 11 sensors-22-09823-f011:**
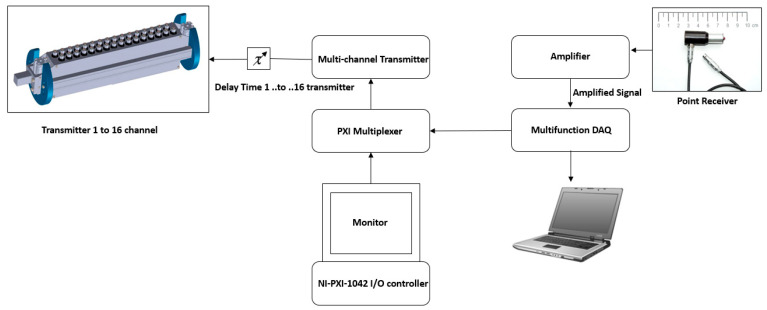
Data Acquisition of third generation borehole probe for directivity measurement.

**Figure 12 sensors-22-09823-f012:**
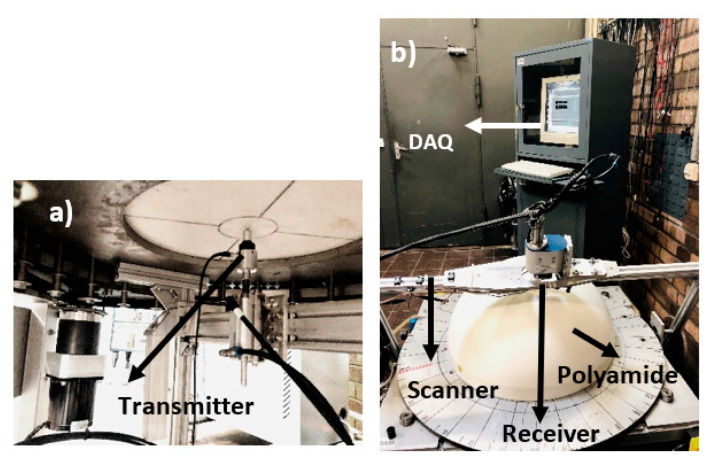
(**a**) Transmitter fixed at bottom flat surface (**b**) Measurement setup.

**Figure 13 sensors-22-09823-f013:**
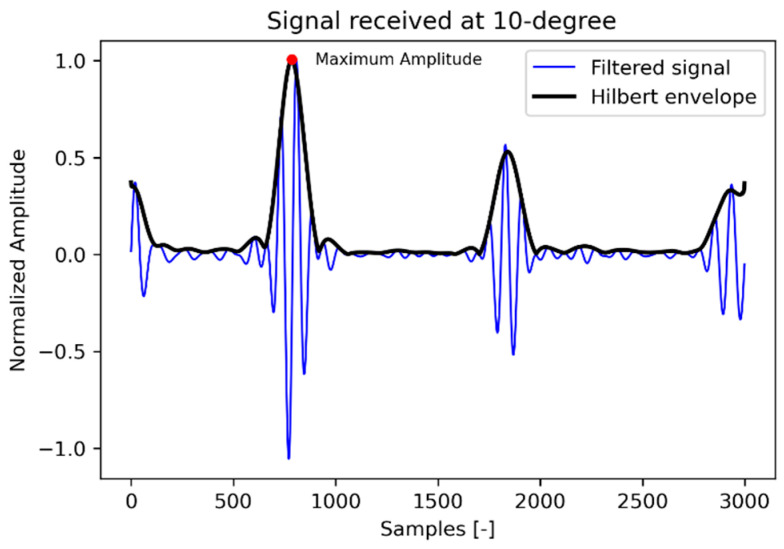
Signal received in polyamide hemispherical test specimen.

**Figure 14 sensors-22-09823-f014:**
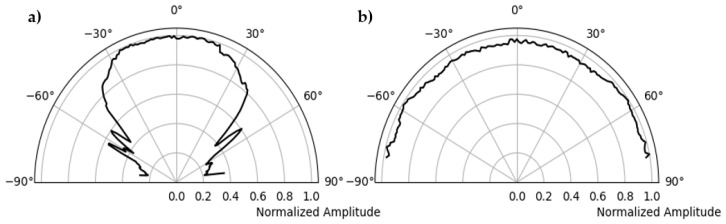
Directivity measurement of point source, (**a**) radiation pattern of single shear wave probe polarization along test plane, (**b**) polarization perpendicular to test plane.

**Figure 15 sensors-22-09823-f015:**
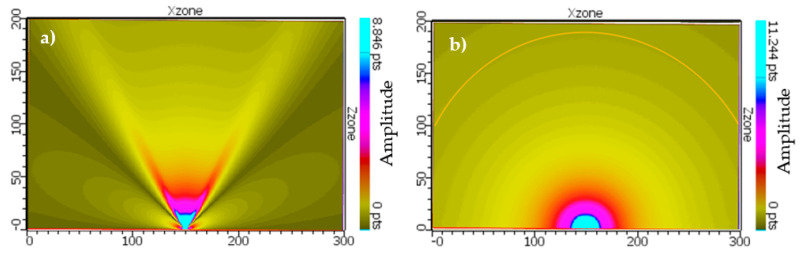
CIVA Simulation of point source, (**a**) radiation pattern of single shear wave probe polarization along test plane, (**b**) polarization perpendicular to test plane.

**Figure 16 sensors-22-09823-f016:**
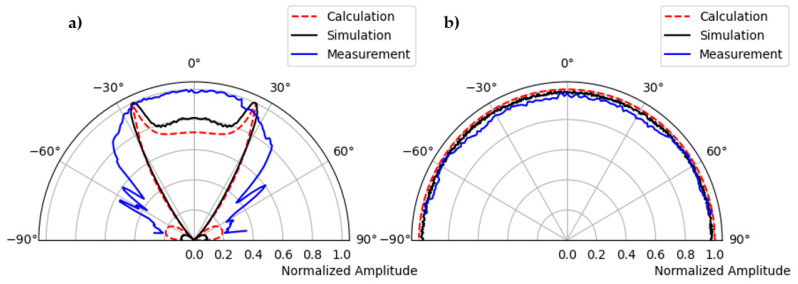
(**a**) Comparison of calculation, simulation, and measurement of radiation pattern of shear horizontal wave, polarization along the test plane, (**b**) polarization perpendicular to the test plane.

**Figure 17 sensors-22-09823-f017:**
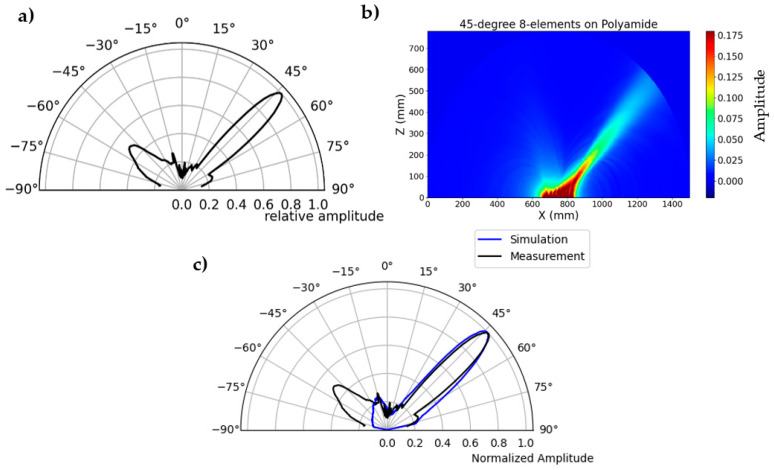
(**a**) Directivity measurement of second generation borehole probe, focused beam at 45°, (**b**) simulation of borehole probe based on measurement parameters, and (**c**) comparison of simulation and measurement.

**Figure 18 sensors-22-09823-f018:**
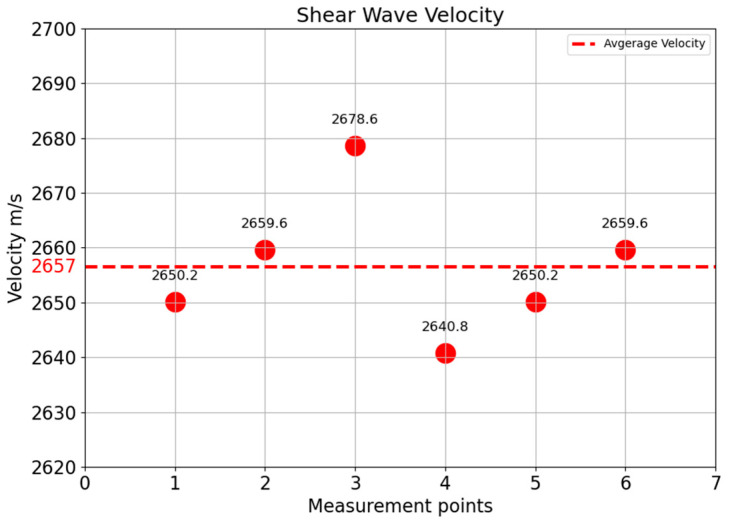
Shear wave velocity of different points on half cylindrical test specimen.

**Figure 19 sensors-22-09823-f019:**
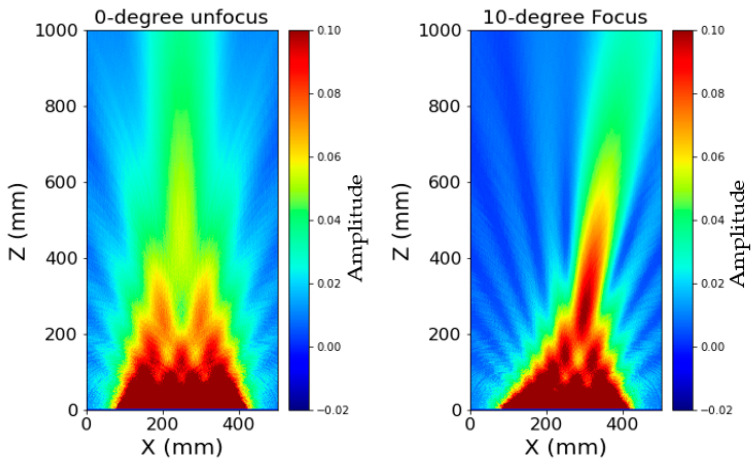
Simulation of unfocused beam and 10° focus beam for third generation borehole probe.

**Figure 20 sensors-22-09823-f020:**
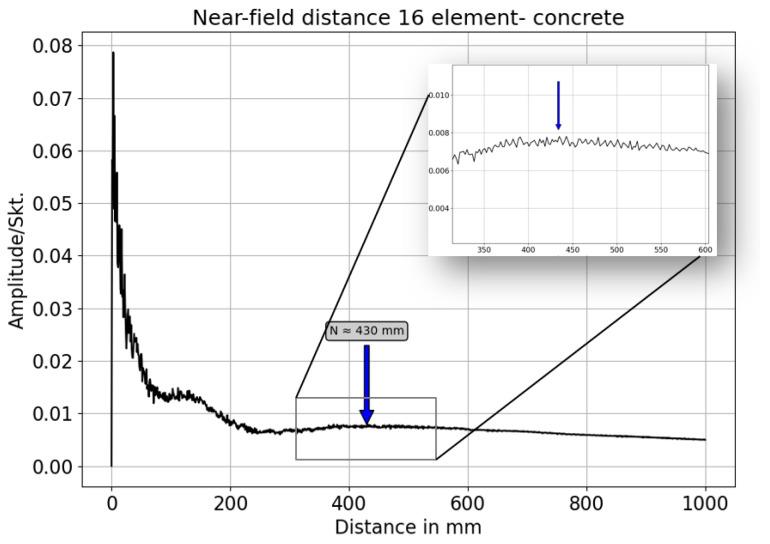
Simulated of near field distance for third generation borehole probe.

**Figure 21 sensors-22-09823-f021:**
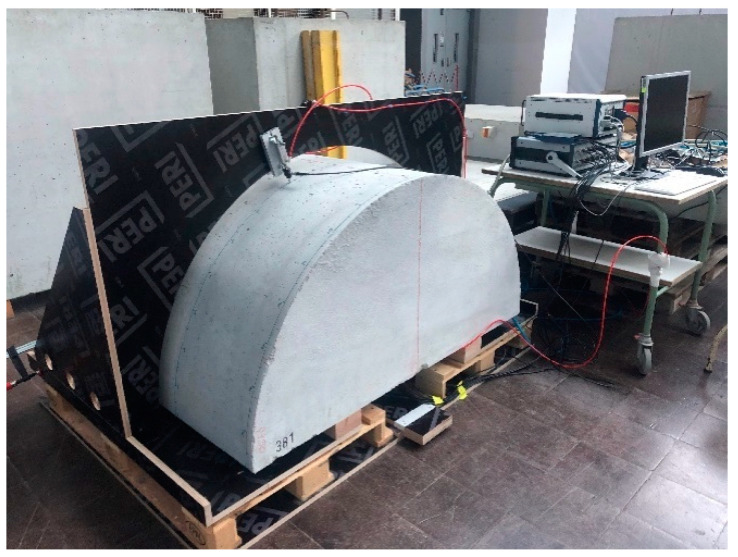
Measurement setup for directivity measurement for third generation probe on concrete half cylindrical specimen.

**Figure 22 sensors-22-09823-f022:**
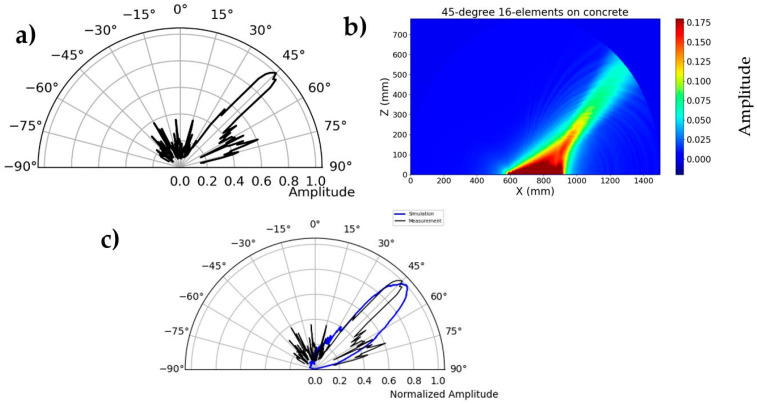
(**a**) Directivity measurement of third generation borehole probe, focused beam at 45°, (**b**) simulation of borehole probe based on measurement parameters, and (**c**) comparison of simulation and measurement.

## Data Availability

The data presented in this study are available on reasonable request from the corresponding author.

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
