# Peer review of "Validation of Novel Ultrasonic Phased Array Borehole Probe by Using Simulation and Measurement"

_sensors, 2022, doi:10.3390/s22249823_

Round 1
Reviewer 1 Report
The research concerns the implementation of a new Ultrasonic Phased Array Borehole Probe. The new probe was developed for in situ quality assurance of sealing structures in radioactive waste repositories using existing research wells. The aim is to examine the salt concrete sealing structures for any cracks and delaminations and to locate the recessed components.
The topic is original and relevant because it concerns the implementation of a new probe for non-destructive examinations on materials. The research offers new tools for non-destructive testing for the evaluation of material properties. The authors showed that the probe prototype increases the signal-to-noise ratio and improves imaging compared to conventional ultrasonic inspection of concrete structures.
Authors should describe in more detail the methodology and results shown in the paper.
- Do not use abbreviation such as i.e. I have seen that you often use this abbreviation, so I will not repeat this advice again, it also applies to the other occurrences.
- Figure 2 must be improved: Add label a and b at two images, not only in the caption
- Authors should emphasize contribution and novelty, the introduction needs to clarify the motivation, challenges, contribution, objectives, and significance/implication.
- You must properly introduce your work, specify well what were the goals you set yourself and how you approached the problem.
- At the end of the section, add an outline of the rest of the paper, in this way the reader will be introduced to the content of the following sections.
- Figure 3 must be improved: Add label a and b at two images, not only in the caption
Section 2 must be improved.
- “phased array technique” Introduce adequately the topic
- Describe in detail the device used in this paper. Extract this data from the datasheet of the instrumentation manufacturer. To make reading the specifications of the instruments more immediate, you can insert them in a table, listing the instruments used and the specific characteristics for each.
- Figure 7 must be improved: Add label a and b at two images, not only in the caption
- “used in by Maack (year) [source]” ? Check
- Replace “test specimen.[14]” with “test specimen [14]”
- Figure 8 must be improved: In the caption the capital letter at the start is missing (polyamide)
- A description of the hardware and software used for data processing is completely missing. Describe in detail the hardware used: Extract this data from the datasheet of the hardware manufacturer. To make reading the specifications of the hardware more immediate, you can insert them in a table, listing the instruments used and the specific characteristics for each. Also, you should describe in detail the software platform you used.
- You must properly introduce the equation, list in detail the variables contained in it with a concise description of the meaning. To make them more readable show them in a bulleted list. In this way the reader will be able to understand the contribution of each variable.
Section 3 must be improved.
- In this section you report the results of your experiments. You should begin the section by describing the experiment set-up in detail.
- Also, you should add photos of the experimental set-up with all the devices connected to the concrete specimens.
- You should also describe in detail how you prepared the concrete specimen by adding all the technical information, characteristics of the ingredients and curing time.
- Figure 12 must be improved: Add label a and b at two images, not only in the caption
- Figure 13 must be improved: Add label a and b at two images, not only in the caption
- Figure 14 must be improved: Add label a and b at two images, not only in the caption
Section 4 must be improved.
- In this section you discuss the results of your experiments. Since the section is very short, it would be advisable to add this discussion to the end of the previous one and rename it to Results and Discussion.
Author Response
Dear Reviewer,
Please find the attached response to the comments.
Sincerely,
Prathik Prabhakara

Reviewer 2 Report
This study studied the orientation characteristics of the ultrasonic beam hole probe array through simulation and experiments. Compared with the second generation, the third -generation instrument increases the number of probes, resulting in huge sound pressure penetration. However, compared to the second generation, the novelty of the third -generation drilling probe only increases the number of elements and the installation of the amplifier, so the innovation of the paper is not obvious enough. It is recommended to increase the application case of this array probe and analyze the advantages of the third generation.
Author Response

(The authors gave the same response as above.)

Round 2
Reviewer 1 Report
The authors addressed all the reviewer's comments with sufficient attention and modified the paper consistently with the suggestions provided. The new version of the paper has improved significantly both in the presentation that is now much more accessible even by a reader not expert in the sector, and in the contents that now appear much more incisive.
Minor revision
- Check the format of equation 1
- In Figure 3 remove the line at the bottom
- Move the two tables in Appendix A in the sections
- Check the format of the references
- Check the format of equation 5 and 6
- In Figure 11 remove the line at the bottom
- In Figure 12 remove the line at the bottom
Author Response
Dear Reviewer2,
Thank you very much for the comments and suggestions.
regards
Prathik

Reviewer 2 Report
No comments or suggestions.
Author Response
Dear Reviewer2,
updated the manuscript, is there any further suggestions? If the English language is not acceptable means I will go for editorial service externally.
Regards
Prathik